# Sulforaphane Reduces Prostate Cancer Cell Growth and Proliferation In Vitro by Modulating the Cdk-Cyclin Axis and Expression of the CD44 Variants 4, 5, and 7

**DOI:** 10.3390/ijms21228724

**Published:** 2020-11-18

**Authors:** Jochen Rutz, Sarah Thaler, Sebastian Maxeiner, Felix K.-H. Chun, Roman A. Blaheta

**Affiliations:** Department of Urology, Goethe-University, 60323 Frankfurt am Main, Germany; Jochen.Rutz@kgu.de (J.R.); gmath@web.de (S.T.); Sebastian.Maxeiner@kgu.de (S.M.); Felix.Chun@kgu.de (F.K.-H.C.)

**Keywords:** sulforaphane, prostate cancer, growth, proliferation, CD44

## Abstract

Prostate cancer patients whose tumors develop resistance to conventional treatment often turn to natural, plant-derived products, one of which is sulforaphane (SFN). This study was designed to determine whether anti-tumor properties of SFN, identified in other tumor entities, are also evident in cultivated DU145 and PC3 prostate cancer cells. The cells were incubated with SFN (1–20 µM) and tumor cell growth and proliferative activity were evaluated. Having found a considerable anti-growth, anti-proliferative, and anti-clonogenic influence of SFN on both prostate cancer cell lines, further investigation into possible mechanisms of action were performed by evaluating the cell cycle phases and cell-cycle-regulating proteins. SFN induced a cell cycle arrest at the S- and G2/M-phase in both DU145 and PC3 cells. Elevation of histone H3 and H4 acetylation was also evident in both cell lines following SFN exposure. However, alterations occurring in the Cdk-cyclin axis, modification of the p19 and p27 proteins and changes in CD44v4, v5, and v7 expression because of SFN exposure differed in the two cell lines. SFN, therefore, does exert anti-tumor properties on these two prostate cancer cell lines by histone acetylation and altering the intracellular signaling cascade, but not through the same molecular mechanisms.

## 1. Introduction

Prostate cancer is the second most frequent malignancy occurring in men worldwide. Based on global cancer statistics (GLOBOCAN), nearly 1,300,000 new prostate cancer cases and 360,000 resulting deaths were recorded in 2018 [1]. Once metastasized, the disease is difficult to treat. Androgen deprivation therapy including surgical or medical castration guarantees at least partial remission. However, tumors inevitably develop resistance to chemotherapy after 12–18 months and metastatic castration-resistant prostate cancer (mCRPC) materializes [2]. Although the pharmacologic armamentarium has grown during the last years with approval of new agents such as abiraterone, enzalutamide, and cabazitaxel, treatment results are still disappointing. Clinical studies point to only moderate improvement in median overall survival of mCRPC-patients treated with abiraterone or enzalutamide compared to placebo, i.e., 15.8 vs. 11.2 months for abiraterone, and 18.4 vs. 13.6 months for enzalutamide [3]. Recent approaches have concentrated on the use of immunotherapeutic agents, programmed cell death 1 (PD-1), PD-ligand 1 (PD-L1), and cytotoxic T-lymphocyte-associated antigen 4 (CTLA-4) antagonists. Still, these strategies are limited to a small subset of patients and it is not yet clear whether immunotherapy actually leads to a significant improvement in patient outcome [4].

During the past decades, complementary and alternative medicine (CAM) has received increasing attention from cancer patients. The main reasons leading to the search for “medical products and practices that are not part of standard medical care” [5] are to boost the immune system, to actively support conventional therapy, to lower the therapeutic side effects, and to diminish the risk of cancer relapse [6,7]. More than 50% of prostate cancer patients are reported to use CAM [8,9], whereby the application of plant-derived compounds is among the most commonly practiced variety of CAM [10,11]. 

Preclinical and clinical investigations have identified a beneficial effect of the natural isothiocyanate sulforaphane (SFN) in treating cancer patients. SFN is highly enriched in its precursor form, glucosinolate, in cruciferous vegetables from the Brassicaceae family such as broccoli, cauliflower, and cabbage. Studies in the United States on a diet rich in broccoli have shown evidence supporting the protective effect of Brassica vegetables against prostate cancer [12]. SFN has recently been declared one of the "Big Five" phytochemicals targeting cancer stem cells [13]. 

SFN acts as a histone deacetylase (HDAC) inhibitor. This is clinically relevant, since epigenetic modification through increased HDAC-activity is involved in tumorigenesis and cancer progression. Because of the reversible nature of epigenetic alteration, the use of HDAC-inhibitors could provide the means to induce malignant cells to revert to their original, healthy, epithelial phenotype. The possibility that SFN as a natural HDAC inhibitor might prevent, delay, or counteract epigenetic alterations in cancer through consumption of an SFN-enriched diet is unquestionably attractive. Nevertheless, the role of SFN in treating prostate cancer is not yet clear and available clinical studies are discussed controversially. This study was designed to investigate SFN’s effect on growth and proliferation of the androgen-resistant tumor cells, PC3 and DU-145. The long-term aim is to implement rationale for a founded treatment option with a compound of natural origin for patients with prostate cancer.

## 2. Results

### 2.1. SFN Blocks Tumor Cell Growth and Proliferation 

Cell growth of both PC3 and DU145 cells was dose-dependently inhibited by SFN in a dose range of 1–20 µM. Significant differences to controls were already apparent at 1 µM SFN (Figure 1A). Cell growth was completely blocked or even reduced below the 24-h value (DU145) with 15 and 20 µM SFN. Therefore, all further investigations were done with 1, 5, and 10 µM SFN. The solvent alone (ethanol, 1:5000 diluted in cell culture medium) did not influence the tumor cell growth (data not shown). Cell viability was similar between treated and non-treated cells, and no significant apoptotic events were apparent at these concentrations, compared to the untreated controls (Appendix A). SFN exposure at 5 or 10 µM also inhibited the proliferation in both cell lines after 24, 48, and 72 h incubation (Figure 1B). Already 1 µM SFN was sufficient to suppress proliferation after 48 or 72 h of incubation in DU145 cells.

### 2.2. Influence of SFN on Clonogenic Tumor Growth

After 10 days exposure to SFN, a dose-dependent reduction of the clone number was apparent, with 1 µM SFN reducing the clone number by approximately half in both PC3 and DU145 cells. About 10 µM SFN led to the complete destruction of all clones (Figure 2A). Microscopic evaluation shows clone disaggregation, already evident with 1 µM SFN (Figure 2B). 

### 2.3. Cell Cycle Evaluation 

Over the course of incubation at 24, 48, and 72 h with SFN (5 or 10 µM), significant shifts in the cell cycle phases in both PC3 and DU145 cells were observed. An initial decrease in the number of G2/M-phase cells with an increase of S-phase cells after 24 h incubation was followed by a progressive elevation in G2/M and loss of G0/G1 phase cells after 48 and 72 h. These effects were stronger with 10 µM SFN than with 5 µM SFN (Figure 3, Appendix A).

### 2.4. Cell-Cycle-Regulating Proteins

Since the cell cycle phases began to show characteristics pointing to anti-tumor activity 48 h into incubation with SFN, cell-cycle-regulating proteins were evaluated following 48 h exposure to SFN (Figure 4, Appendix A). Although SFN had the same kind of influence on cell phase distribution in the DU145 and PC3 cell lines, the cell-cycle-regulating proteins in the two cell lines were different. With 10 µM SFN, CDK1 and CDK2 increased in both cell lines, whereas the respective ligands cyclin A and B were only enhanced in the PC3 cells (each compared to the untreated controls). Contrary to the elevation of the total CDK 1 and 2 proteins, SFN at 5 µM (pCDK1) or at 5 and 10 µM SFN (pCDK2) induced a significant suppression of the respective protein phosphorylation in PC3 cells. Acetylation of histones H3 and H4 as well as p19 increased in both PC3 and DU145 cells. SFN displayed an opposing regulatory mechanism in regard to p27, which was up-regulated in DU145, but down-regulated in PC3 cells. 

Pilot studies were carried out to explore the expression of the CD44 variants v3-v10 on prostate cancer cells. Since CD44v4, v5, and v7 expression levels were greatest, the effect of SFN on these particular CD44 types was investigated. The basal expression of CD44v4, v5, and v7 at zero time is shown in Figure 5A (DU145) and Figure 6A (PC3). SFN applied at 5 and 10 µM induced an increase in the CD44 variants on both DU145 and PC3 (Figure 5B and Figure 6B, respectively). However, the effect of SFN was time-dependent. SFN incubation for 24 h evoked an up-regulation of CD44v4 only on the PC3 cells. After 48 h incubation, all CD44 variants were up-regulated on PC3 cells, whereas the same exposure to SFN only up-regulated CD44v5 on DU145 cells. The most prominent alterations were seen after 72 h SFN exposure with a doubling of CD44 v4, v5, and v7 on both the DU145 and PC3 cell lines, all compared to the respective controls. 

### 2.5. CD44 Blockade

Since a stronger inhibiting effect of SFN on growth and proliferation in DU145 than in PC3 cells was apparent, CD44 blocking studies were carried out on DU145 cells to examine the role the CD44 variants may play on clonogenic activity. Under control conditions, 204 +/− 28 clones were counted (*n* = 3). The clone number was significantly elevated by 26.5% to 258 +/− 34 clones in the presence of CD44v4, v5 and v7 monoclonal antibodies.

## 3. Discussion

SFN suppressed the growth and proliferation of PC3 and DU145 cells in vitro. Distinct effects were apparent with 1 µM SFN, whereby the sensitivity to SFN depended on the cell line and whether growth or proliferation was assayed. About 5 µM SFN already reduced the tumor cell number of PC3 and DU145 by about 50% when evaluated after 72 h. However, to achieve a 50% reduction in DU145 cell proliferation, treatment with 10 µM SFN was necessary. Total of 10 µM SFN reduced PC3 proliferation to a lesser extent, compared to DU145, indicating that the efficiency of SFN on tumor proliferation may depend on the cell line. The discrepancy in cell line sensitivity also became evident when measuring the clonogenic growth, where DU145 cells responded more strongly to SFN than PC3 cells. Other investigators have calculated the IC50 value for SFN to be 14 µM for DU145 versus 32 µM for PC3 [14]. Geno-and phenotypic characterization indicate distinct differences between these two cell lines [15], so that patient response to SFN could depend on the genomic cancer signature. 

Micromolar plasma concentrations of SFN have been shown to exert distinct anti-tumor effects, making the SFN dosage used in the present investigation relevant. Four hours after the dogs had consumed 24 mg SFN, plasma concentrations of SFN and SFN metabolites peaked at 0.004–0.095 µM, followed by a significant 25% decrease of HDAC activity in peripheral blood mononuclear cells 24 h post-consumption [16]. A transient decrease in HDAC activity has also been observed in healthy humans 3 h after providing a daily 200 µM SFN dose, resulting in a plasma concentration of SFN metabolites of 0.1–0.2 µM [17]. Administration of 200 µM SFN to melanoma patients resulted in a significant reduction of pro-inflammatory cytokines [18], also observed in another trial where patients received SFN at a daily dose of 2.2 µM/kg body weight, with a mean plasma level of 0.13 µM [19]. These investigations point to biochemical activity of SFN in the micromolar range, indicating that the concentrations applied in the present in vitro investigation could be clinically relevant. 

SFN (5 and 10 µM) led to an accumulation of PC3 and DU145 cells in the G2/M-and S-phase of the cell cycle after 48 and 72 h. About 5 and 10 µM SFN concentrations were similarly effective in DU145 cells, whereby 10 µM SFN was more effective than 5 µM in PC3 cells. In accordance, Myzak et al. observed an increased number of PC3 in the G2/M phase following 48-h exposure with 15 µM SFN [20]. There is also evidence of a time-dependent cell cycle modulation. Clarke et al. noted a cell cycle arrest of PC3 in G2/M 24 h after incubation with 15 µM SFN, but arrest in the S-phase only after 48 h [21]. Since our results show an increase in S-phase cells, paralleled by a decrease in G2/M-phase cells after 24 h SFN exposure, the kind of cell cycle blockade might indeed depend on the drug exposure time. The percentage of PC3 cells in the S-phase has been shown to increase after 6 h of treatment with SFN, whereas the fraction of cells in G2/M increased after 16 h [22]. Based on these observations, it may be assumed that SFN diminishes PC3 and DU145 cell proliferation by causing both a G2/M- and S-phase block. 

Differences between DU145 and PC3 cells in regard to the influence of SFN on cell-cycle-regulating proteins were apparent. Cdk1 and 2 (total) were elevated by SFN in DU145 cells but both Cdk 1 and 2 and their respective binding partners, cyclin B and A, were enhanced in PC3 cells. The role of the Cdk-cyclin axis in cancer remains unclear. Osteosarcoma cells displayed a G2/M phase arrest under SFN influence that was associated with decreased cyclin A and B, along with Cdk1 and 2 [23]. In another investigation with osteosarcoma cells, the induction of a G2/M phase arrest by SFN correlated with a Cdk1 increase [24]. SFN also induced a G2/M phase arrest in colon cancer cells, accompanied by an elevation of cyclin A, cyclin B, and Cdk2, but decreased Cdk1 protein expression [25]. Other investigators, however, have observed an SFN-induced up-regulation of Cdk1 in colon cancer-derived tumors [26]. An SFN-induced mitotic arrest with an induction of cyclin B and increased cyclin B-Cdk1 complex has also been shown in a bladder cancer cell model [27]. The elevation of Cdk1 and 2 in DU145 cells and cyclin A and B in PC3 cells may be the respective drivers from the S into the G2/M phase. Although Cdk1 and 2 were enhanced by SFN in PC3 cells the phosphorylated Cdk1 and 2 proteins were reduced. Suppression of the active Cdks could mitigate the role of Cdk1 and 2 in promoting the G2/M entry of PC3 cells and, therefore, explain why this cell line is less sensitive to SFN than the DU145 cell line. 

p19 plays a key role in cell cycle blockage and apoptosis and is considered a potent tumor suppressor. It is connected to the opposite-acting Akt pathway and inhibition of Akt signaling is associated with increased expression of p19 [28]. SFN significantly up-regulated p19 in the DU145 (5 and 10 µM SFN) and PC3 (10 µM SFN) cells. Recent experiments with renal cell carcinoma cells have shown that an SFN-induced p19 increase is coupled to Akt deactivation [29]. Particularly HDAC-inhibitors have been identified as having strong potential in targeting p19 [30]. We have found that SFN induces an increase in the acetylated histones H3 and H4 in DU145 cells and, to a lesser extent, in PC3 cells as well. These findings reinforce SFN’s functioning as a natural HDAC-inhibitor. 

Whether p27 is as epigenetically controlled as p19 cannot be answered with certainty. Elevated H3 acetylation levels in the VCaP (human prostate cancer) cell line has been shown to correspond to p27 enhancement [31]. This effect was seen in DU145 cells, but not in PC3 cells where p27 was down-regulated by SFN. Five µM SFN did not induce aH4 and 10 µM SFN did not up-regulate aH3 in PC3 cells. This indicates that p27 plays different regulatory roles in the two cell lines. p27 expression can also depend on the phosphorylation level of Cdk1 [32]. Indeed, pCdk1 as well as pCdk2 were suppressed by SFN in the PC3 cells. Speculatively, feedback between pCdk1 and p27 could lead to blockage of the cell cycle progression in PC3 cells. This, however, is speculative and requires verification by further studies. 

Conflicting results in regard to CD44 expression and tumor behavior have been reported. Several reports point to a positive association between CD44v expression and tumor progression, whereas others find no relationship or even a negative association between CD44v and tumor growth and metastasis. The present investigation documents a significant elevation of CD44v4, v5, and v7 by SFN on the DU145 and PC3 cell surface. Since SFN blocked the tumor growth and proliferation, it is likely that these CD44 variants contributed to this process. Indeed, the blocking study done with DU145 cells displayed an inverse correlation between tumor proliferation (clonogenic growth) and the CD44v4, v5, and v7 expression levels. 

With respect to these particular subtypes, CD44v4 reduction has been observed during breast cancer development and progression [33]. Immunohistochemical analysis of CD44v5 expression in radical prostatectomy specimens has revealed decreased CD44 scores during de-differentiation from low- to high-grade prostatic intraepithelial neoplasia [34]. Loss of CD44v5 has also been documented in tumor cells isolated from a lymph node metastasis of a poorly differentiated carcinoma [35] and in bladder tumor tissues [36]. Examination of cribriform prostate cancer by immunohistochemistry demonstrated an inverse correlation between proliferative activity and CD44v7/8 expression [37]. Accordingly, CD44v7 was significantly down-regulated in primary squamous cell carcinomas and was not detectable in the majority of metastasis-derived specimens [38]. Based on a recent publication, exposure of SFN to bladder cancer cells induced a significant increase of CD44v4 and v7 and diminished chemotaxis, whereby knock-down of CD44 correlated with enhanced chemotaxis [39]. This broad spectrum of effects of SFN on different tumor entities indeed shows that SFN exerts its influence through different molecular mechanisms, depending on the tumor type.

Disparity between effects caused by SFN on DU145 and PC3 cell-cycle-regulating protein expression may thus be explained. The cell cycle regulator AKT is differently modified in PC3 and DU145 cells in the presence of the HDAC-inhibitor valproic acid [40]. Valproic acid has also been shown to modify the adhesion behavior of PC3 and LNCaP cells differently, and has been traced back to differing integrin equipment [41]. The disparate mechanistic influence of SFN has also been verified on several bladder cancer cell lines, with the supposition that SFN acts on a set of integrin receptors that may differ according to the initial characteristic integrin composition of the particular cell type [39]. We did not investigate whether this may hold true in the present investigation. However, cross-talk between integrins and CD44 has been documented [42,43], making it likely that CD44 variants are connected to different integrin sets, finally leading to diversely activated intracellular signaling. Still, this is purely speculative and not supported by sound data.

Although the role of CD44v in tumor biology requires further investigation, it may be assumed that up-regulation of CD44v4, v5, and v7 is crucial to SFN’s slowing proliferative activity in the prostate cancer cells examined in the present study. Zhang et al. postulated that this may occur by re-switching a mesenchymal tumor cell phenotype to an epithelial phenotype [44]. This has been confirmed by other investigators who have demonstrated that CD44 variant splice isoforms are important drivers for a mesenchymal-epithelial transition [45]. Recently, the expression of CD44v in DU145 cells has been shown to positively correlate with E-cadherin and negatively correlate with cell migration and invasion [46]. Since epithelial/mesenchymal markers were not evaluated in the present investigation it cannot be concluded with certainty that a similar scenario took place. Still, SFN has been shown to increase E-cadherin and decrease N-cadherin and vimentin expression in endometrial cancer cells. This was associated with a G2/M cell cycle arrest [47]. An increase in G2/M-cells under SFN was also seen in the present investigation. Since loss of CD44v correlated with enhanced clonogenic growth, it is likely that CD44v directly contributed to the cell cycle blockade in our model system. Aside from cell growth regulation, CD44v may also be involved in prostate cancer cell invasion [46]. Whether SFN regulates metastatic progression of prostate cancer cells via CD44v requires further investigation. 

## 4. Materials and Methods 

### 4.1. Cell Culture

Human, castration-resistant prostate tumor cell lines PC3, DU-145, obtained from DSMZ (Braunschweig, Germany) were grown and subcultured in RPMI 1640 medium (Gibco/Invitrogen, Karlsruhe, Germany) augmented with 10% fetal calf serum (FCS), 2% HEPES (2-[4-(2-hydroxyethyl)piperazin-1-yl]ethanesulfonic acid) buffer (1 M, pH 7.4), 2% glutamine, 1% penicillin/streptomycin at 37 °C in a humidified, 5% CO_2_ incubator. 

### 4.2. Sulforaphane

SFN (L-Sulforaphane, Biomol, Hamburg, Germany) ranging from 1–20 μM was applied to cell cultures and controls remained untreated. Toxic effects of SFN, reflected by cell viability were checked with trypan blue (Gibco/Invitrogen, Darmstadt, Germany). 

### 4.3. Tumor Cell Growth 

Cell growth was assessed using the 3-(4,5-dimethylthiazol-2-yl)-2,5-diphenyltetrazolium bromide (MTT) dye reduction assay (Roche Diagnostics, Penzberg, Germany). The tumor cells (100 µL, 1 × 104 cells/mL) were seeded onto 96-well tissue culture plates and then treated with SFN (1–20 μM). SFN remained in the culture medium and was not removed. Controls were incubated without SFN. After 24, 48, and 72 h, MTT (0.5 mg/mL) was added for an additional 4 h. Thereafter, cells were lysed in a buffer containing 10% SDS in 0.01 M HCl. The plates were incubated overnight at 37 °C, 5% CO_2_. Absorbance at 550 nm was determined for each well using a microplate ELISA reader. Defined numbers of cells (in triplicate) ranging from 2500–160,000/well were added to the microtiter plates to correlate absorbance with cell number. Results were expressed as mean cell number after subtracting the background absorbance from cell culture medium alone.

### 4.4. Apoptosis

The influence of apoptosis in regard to tumor cell growth was assessed with the Annexin V-FITC Apoptosis Detection kit (BD Pharmingen, Heidelberg, Germany) that quantifies binding of Annexin V/propidium iodide. Tumor cells, incubated with SFN (controls were without SFN) were washed twice with PBS, and incubated with 5 μL Annexin V-FITC and 5 μL propidium iodide in the dark for 15 min at room temperature. Cell numbers were determined by flow cytometry (FACScalibur, BD Biosciences, Heidelberg, Germany). Percentages of early and late apoptotic, necrotic and vital cells in each quadrant were calculated using CellQuest software (BD Biosciences).

### 4.5. Tumor Cell Proliferation

Cell proliferation was measured using a BrdU (Bromodeoxyuridine/5-bromo-2’-deoxyuridine) cell proliferation enzyme-linked immunosorbent assay (ELISA) kit (Calbiochem/Merck Biosciences, Darmstadt, Germany). PC3 or DU145 cells, seeded onto 96-well microtiter plates, were incubated with 20 µL BrdU-labeling solution per well for 8 h, fixed and detected using anti-BrdU mAb. Absorbance (optical density, OD) was measured at 450 nm. Proliferation was measured after 24, 48, and 72 h in the presence of SFN (1, 5, 10 µM) or cell culture medium alone (controls). 

### 4.6. Clonogenic Growth Assay 

500 single PC3 or DU145 cells (treated with 1, 5, or 10 µM SFN versus non-treated) were transferred to 6-well plates. Following 10 days incubation without medium change, cell colonies were fixed and counted. Clones of at least 50 cells were counted as one colony.

### 4.7. Cell Cycle Analysis

Cell cycle analysis was carried out on subconfluent cell cultures. Tumor cell populations, treated with 5 or 10 µM SFN (controls were not treated) were stained with propidium iodide, using a Cycle TEST PLUS DNA Reagent Kit (BD Biosciences) and then subjected to flow cytometry (FACScalibur flow cytometer, BD Biosciences). 10,000 events were collected from each sample and data were acquired using Cell-Quest software. ModFit software (BD Biosciences) was used to assess cell cycle distribution. The number of gated cells in G0/G1-, G2/M-, or S-phase was presented as % of total cells.

### 4.8. Western Blot Analysis

Cell-cycle-regulating proteins were investigated in DU145 and PC3 cells treated with 5 or 10 µM SFN (controls remained untreated). After applying tumor cell lysates to a 7% polyacrylamide gel they were electrophoresed for 90 min at 100 V. Protein was then transferred to nitrocellulose membranes (1 h, 100 V), blocked with non-fat dry milk for 1 h, and the membranes were then incubated overnight with the following monoclonal antibodies: CDK1 (IgG1, clone 1), pCDK1/Cdc2 (pY15; IgG1, clone 44/Cdk1/Cdc2), CDK2 (IgG2a, clone 55), cyclin A (IgG1, clone 25), cyclin B (IgG1, clone 18), p19 (IgG1, clone 52), Kip1/p27 (IgG1, clone 57); all from BD Biosciences, pCDK2 (Thr160; Cell Signaling). Histone acetylation was investigated by marking with anti-acetylated H3 (IgG, clone Y28) and anti-acetylated H4 (Lys8, polyclonal IgG); all from Cell Signaling. HRP-conjugated goat anti-mouse IgG and HRP-conjugated goat anti-rabbit IgG (Cell Signaling) served as the secondary antibody. To visualize the proteins the membranes were briefly incubated with ECL detection reagent (Amersham/GE Healthcare, München, Germany) and then analyzed by the Fusion FX7 (Peqlab, Erlangen, Germany). The internal control was β-actin (Cell Signaling). Pixel density analysis of the protein bands (both total and phosphorylated) and calculating the ratio of protein intensity/β-actin intensity was carried out with GIMP 2.8 software.

### 4.9. CD44 Expression

Conjugating the CD44 variants 44v4, v5, and v7 antibodies was carried out with the Lightning-Link Allophycocyanin (APC) Conjugation Kit (eBioscience, ThermoFisher, Darmstadt, Germany). After detachment and washing DU145 or PC3 cells with blocking solution (PBS, 0.5% BSA) cells were then incubated for 1 h at 4 °C with 2.5 µL APC-conjugated monoclonal antibody directed against the following CD44 variants: anti-CD44v4 (clone VFF-11), anti-CD44v5 (clone VFF-8), and anti-CD44v7 (clone VFF-9; all: Bio-Rad, Feldkirchen, Germany). Total of 5 µL APC mouse IgG1, K (clone P3.6.2.8.1; ThermoFisher, Dreieich, Germany) served as the control isotype. CD44 expression was then assessed employing a FACscan (BD Biosciences; FL4-H (log) channel histogram analysis; 1 × 104 cells per scan) and expressed as mean fluorescence units (MFU). 

### 4.10. Blocking Studies

To investigate how CD44v contributes to tumor cell proliferation, DU145 cells were treated with sodium hyaluronate (MW 500–750 kDa; Sigma Aldrich, München, Germany) and with an anti-CD44 monoclonal antibody cocktail consisting of anti CD44v4, v5, and v7 (10 ng/mL each). Controls remained untreated. Tumor cells were then subjected to the clonogenic growth assay. 

### 4.11. Statistics

All experiments were performed 3–6 times and statistical significance was determined with the Wilcoxon-Mann-Whitney-U-test or Student’s t-test. Differences were considered statistically significant at *p* < 0.05. 

## 5. Conclusions

SFN up-regulated CD44v4, v5, and v7, slowing the proliferative activity in DU145 and PC3 tumor cells. p19, a potent tumor suppressor, playing a key role in cell cycle blockage and apoptosis, was also up-regulated. SFN application drove the prostate cancer cells from the S into the G2/M cell cycle phase. These findings indicate that application of the natural compound, SFN, could improve the current prostate cancer treatment protocols. Since studies on humans are sparse, ongoing trials are necessary to evaluate whether SFN can prove clinically beneficial. SFN dosage, bioavailability, optimal indication, and potential toxic effects are further issues that require attention.

## Figures and Tables

**Figure 1 ijms-21-08724-f001:**
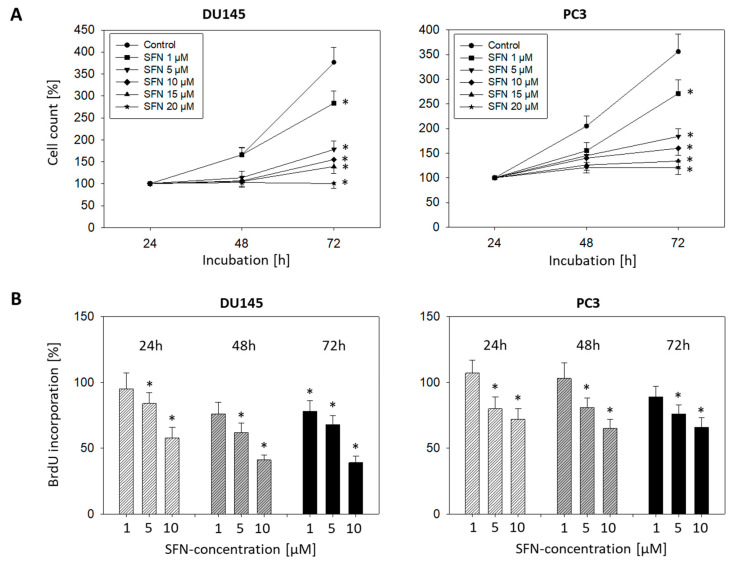
(**A**): Influence of 1–20 µM sulforaphane (SFN) on cell growth of DU145 and PC3 cells. Cell number evaluated after 24 (100%), 48, and 72 h by MTT assay. Whiskers indicate standard deviation, * = *p* ≤ 0.05, *n* = 6. (**B**) Influence of 1, 5, and 10 µM SFN on proliferation of DU145 and PC3 cells. Evaluation by BrdU incorporation after 24, 48, and 72h. Whiskers indicate standard deviation. * Indicates significant difference to untreated controls set to 100%, *p* ≤ 0.05. *n* = 3.

**Figure 2 ijms-21-08724-f002:**
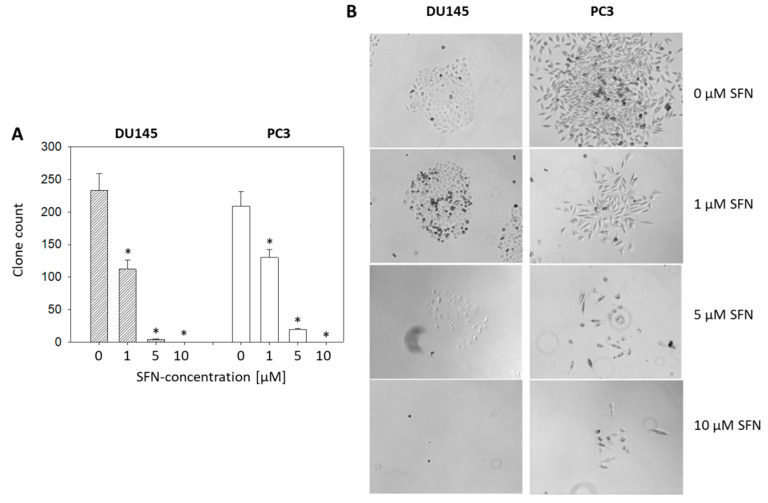
(**A**) Influence of 1, 5, and 10 µM sulforaphane (SFN; 10 day exposure) on clonogenic growth. Whiskers indicate standard deviation. * Indicates significant difference to untreated controls, *p* ≤ 0.05. *n* = 3. (**B**) Morphologic alteration after SFN exposure.

**Figure 3 ijms-21-08724-f003:**
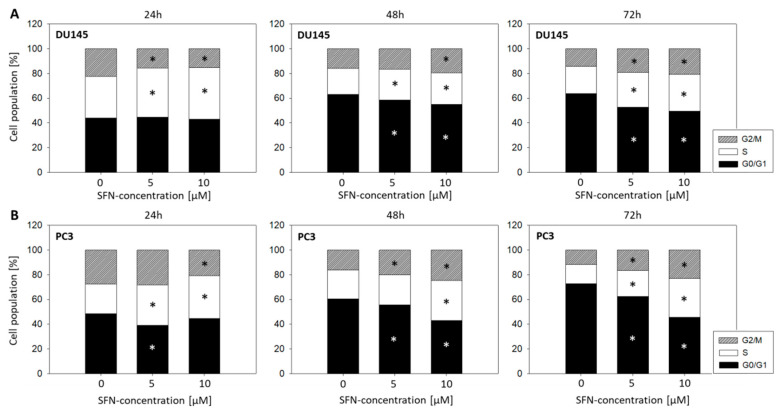
Influence of 5 and 10 µM sulforaphane (SFN) on proportionate G0/G1, S, and G2/M-phases of the cell cycle in DU145 (**A**) and PC3 (**B**) cells over the course of 72 h. (*n* = 3; * indicates significant difference to untreated controls).

**Figure 4 ijms-21-08724-f004:**
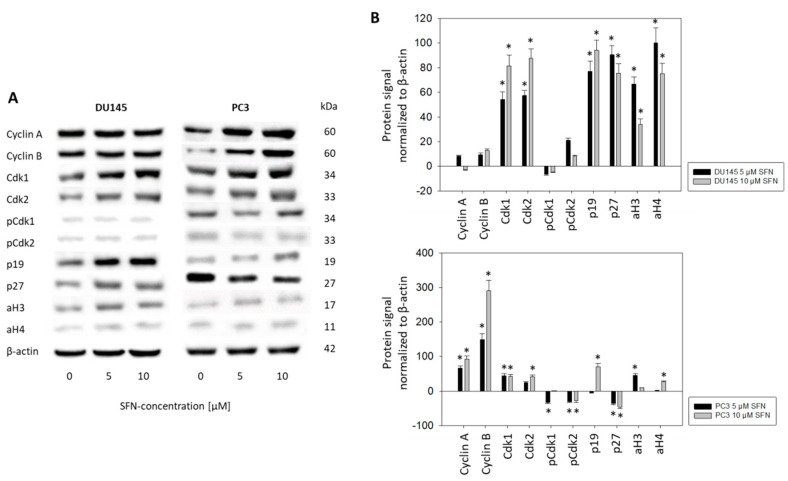
(**A**) Protein profile of cell-cycle-regulating proteins and histone H3 and H4 acetylation after exposure to 5 or 10 µM sulforaphane (SFN). Controls were untreated (0 µM SFN). One representative of three separate experiments is shown. Each protein analysis was accompanied by a β-actin loading control. One representative internal control is shown. (**B**) The ratio of protein intensity/β-actin intensity was calculated and expressed as a percentage of the controls, set to 100%. * Indicates significant difference to controls, *p* ≤ 0.05. *n* = 3.

**Figure 5 ijms-21-08724-f005:**
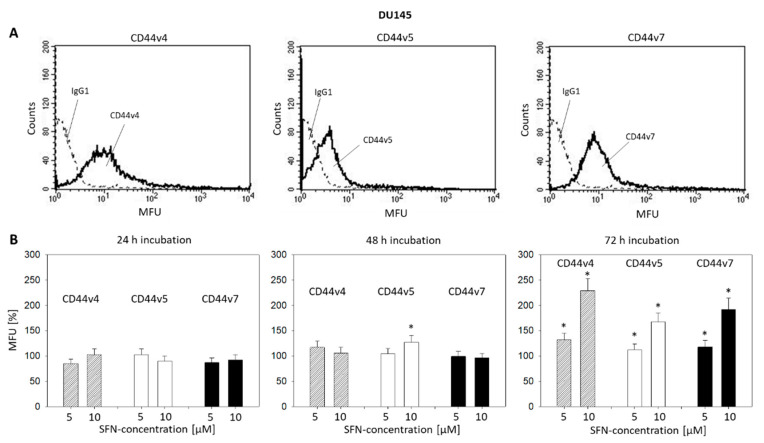
(**A**) CD44 variants v4, v5, and v7 on DU145 cells. MFU = mean fluorescence units. Single representative of three separate experiments. Solid line: specific fluorescence; dashed line: isotype IgG1-APC. (**B**) CD44 variants v4, v5, and v7 following sulforaphane (5 and 10 µM) exposure. Means related to untreated controls (100%). Whiskers indicate SD. * = significant difference to corresponding control, *p* ≤ 0.05, *n* = 4.

**Figure 6 ijms-21-08724-f006:**
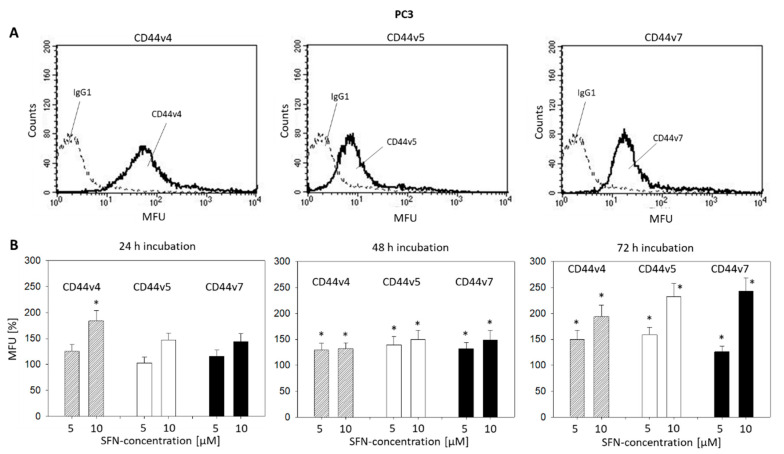
(**A**): CD44 variants v4, v5, and v7 on PC3 cells. MFU = mean fluorescence units. Single representative of three separate experiments. Solid line: specific fluorescence; dashed line: isotype IgG1-APC. (**B**): CD44 variants v4, v5, and v7 following sulforaphane (5 and 10 µM) exposure. Means related to untreated controls (100%). Whiskers indicate SD. * = significant difference to corresponding control, *p* ≤ 0.05, *n* = 4.

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
