# Peer review of "Sulforaphane Reduces Prostate Cancer Cell Growth and Proliferation In Vitro by Modulating the Cdk-Cyclin Axis and Expression of the CD44 Variants 4, 5, and 7"

_ijms, 2020, doi:10.3390/ijms21228724_

Round 1

Reviewer 1 Report

In this manuscript, the authors describe findings demonstrating sulforaphane (SFN), an isothiocyanate present in cruciferous vegetables, which reduces cancer cell growth and proliferation using two types of human prostate cancer cell lines. It has been reported that SFN shows therapeutic effects in several types of cancer in in vitroand animal models, as well as in clinical studies. However, its molecular mechanism is not yet clear. The submitted manuscript demonstrates that the mechanism underlying slowing proliferative activity of prostate cancer cell by SFN seems to be related to cell cycle regulating proteins and expression of CD44 variants. Although the manuscript contains convincing data, it fails to meet the quality standards for publication for the following reasons.

Major comments

  • The reasons why the authors focused on the expression of CD44v4, v5 and v7 out of nine variants known in humans are not described in the manuscript. It is important to mention specific reasons underlying the approach the authors took. As mentioned in the manuscript, it has been reported that SFN-treated DU145 cells showed inhibited expression of CD44v6, which seems to be involved in the aggressive behavior of various types of cancer (Peng et al., Oncology Reports, 2015), implying that SFN differently affects the CD44 variant depending on its targeting variant. As the finding about modulating expression of CD44v4, v5 and v7 by SFN is novel, it is also important to indicate reference publications to support the possible linkage between decreased expression of CD44v4, v5 and v7 and suppression of cancer cell proliferation, which could reinforce the data of CD44 blocking study.
  • It is necessary to perform statistical analysis for the data in cell cycle evaluation (figure3) and in the CD44 blocking study.

Cell cycle evaluation: The authors need to describe the change of cell numbers in each cell cycle by SFN treatment on the basis of statistical values. Showing a modified form of the graph to indicate comparison of cell populations between SFN 0 uM, 5 uM and 10 uM in each cell cycle would be helpful to illustrate this point.

CD44 blocking study: It should be possible to simply perform Student’s-T test for the clonogenic activity between the control and the antibody cocktail-exposed cell.

  • Although the Material and Methods describes Apoptosis analysis, the data is not shown in the manuscript. As evaluation of apoptosis could be critical in the context of manuscript, it is necessary to show the actual data.
  • The Material and Methods mentions the evaluation of the toxic effect of SFN determined by trypan blue; however the data is not included. As this is critical data to determine the treatment dose, it is important to show the data.

Minor comments

  • The authors demonstrated that PC3 and DU145 showed different effects toward expression of cell cycle regulating protein and change of CD44v4, v5 and v7 by SFN, suggesting the molecular action observed in the two cell lines are different. It is highly recommended to discuss the possible factors causing the differential profile between the two cell lines.
  • The authors need to describe the treatment design in detail. It is not clear how many doses were administrated to the cell over the course of incubation. Was the medium, which included SFN, changed during 24 - 72 hours or 10 days of incubation?
  • On Page 4, line 105, the authors describe that cyclin A and B were only enhanced in the DU145. However, according to the data shown in figure 4B, it is assumed that cyclin A and B were elevated in the PC3, not This should be corrected.
  • The images of protein profiles in figure 4A are too small. Please use larger images.
  • The title of the Y-axis in figure4 B could mislead the readers. Although it is assumed that the relative expression of each protein normalized by Beta-actin was expressed by %, it would be better to edit the title of the Y-axis to clearly signify its point.
  • If the authors do not have any supplementary materials, please delete line 339 on page 11.

Reviewer 2 Report

These investigators have pursued the idea that aggressive prostate cancer that is resistant to current conventional therapies might be treatable with phytochemicals like sulforaphane (SFN) showing evidence of potent anti-cancer efficacy. Using the androgen resistant aggressive prostate cancer cell lines, DU145 and PC-3, they conducted a dose and time dependent treatment in vitro with SFN. They analyzed cell proliferation (MTT assay), clonogenic growth (colony formation), and a panel of cell cycle relevant proteins including acetylated histones using flow cytometry and western immunoblot. They noted that higher concentrations of SFN were cytotoxic and therefore restricted studies to a range of 1-10 uM SFN. Since CD44 and variants play important roles in tumor cell progression they analyzed CD44v4,5,7 cell surface expression +/- SFN treatment. Standard statistical analyses were applied.

The results that were convincing demonstrated the following:

Figure 1: SFN even at 1uM showed a significant inhibition of cell proliferation of both cell lines. The inhibition was dose dependent in the 1-10uM range.

Figure 2: SFN significantly and dose dependently inhibited colony formation (clonogenic assay) with a remarkable inhibition at 5uM SFN with DU145 somewhat more sensitive.

Figure 3: After 72h treatments there was a greater reduction of cells in G0/G1 with increase in G2/M suggesting an arrest in G2/M, highest in 10uM SFN.

Figure4: Western blot analysis did show overt increases in pCDK1 for PC-3 and modest increase in pCDK2 with a corresponding significant decrease in p19. DU145 did not show the cDK1/2 effects but p19 was increased. On the other hand, p27 decreased in PC-3 and increased in DU-145. In terms of histone acetylation, modest increases were observed in agreement with previous reports that SFN induces histone acetylation.

Figure5: Surface expressions of CD44v4,5,7 were significantly upregulated in both cell lines where after 72h 10uM SFN treatment there was a significant increase in CD44v4,5 and 7 in DU145 and also in PC-3 to a greater extent. 10uM SFN already had a significant induction of CD44v4 in PC-3.

The conclusion that the two cell lines respond differentially to SFN at a biochemical level but respond similarly to inhibition of proliferation and clonogenic growth is valid. The differences in term so cell cycle responses likely reflect intrinsic differences at a genetic level and overall tumor phenotype, obvious by phase contrast. Although the manuscript is well written and grammatically clean and figures are accurately configured, there are some missing data not shown:

  1. The Annexin-V apoptosis results are not shown in main figures or supplementary? Needs a figure.
  2. The CD44 blockade results were also not shown but a result described? Needs a figure separate or joined.

Other points:

  1. Result 2.1: Is there apoptosis after 72h of treatment in concert with a possible default situation after cell cycle arrest?
  2. Result 2.2: The clonogenic assay assumes growth from single cells. Was this verified from day 0 onward? Otherwise it is a colony forming assay.
  3. Provide the carrier solution for SFN that should have been used as the control.
  4. Figure 4: What is meant by one representative internal control? It would be easier to interpret if 0 SFN values are included in the histograms next to 5 and 10uM.
  5. Figure 6: What were the surface expression levels of CD44 and variants at time 0 for these cell lines? Would like to see any shifts after 5 and 10uM SFN.
  6. Discussion: Discussion is appropriate and informative relative to results obtained.

A] Is it possible that SFN treatment leads to differentiation of these cell lines? Was this looked at and could correlate with cell cycle blocks?

B] Changes in acetylated histones H3 and H4 were low to modest. Is this a function of time or growth form since 3D growth of these cell lines are more representative of the cancer.

 C] Any other data on CD44 parent molecule relative to variants?

As both cell lines are aggressive metastatic lines what role can these CD44v have?

D] Do the investigators have any morphological evidence of EMT/MET after SFN treatment?

Round 2

Reviewer 2 Report

The authors have addressed the concerns and importantly stated the limitations of their study. As conducted, it does show interesting aspects in the expression of CD44variants and effect of SFN on these cell lines. They did acknowledge with added literature that cancer cell lines can respond differentially, an area requiring a lot more study. Their work will certainly have to be further scrutinized and repeated by other investigators. I feel that they did address the issue of controls in a logical manner so no longer an issue.  Therefore, the manuscript is acceptable for publication.